# Matrix Completion From any Given Set of Observations

**Troy Lee**
Nanyang Technological University and
Centre for Quantum Technologies
troyjlee@gmail.com

Adi Shraibman
Department of Computer Science
Tel Aviv-Yaffo Academic College
adi.shribman@gmail.com

## Abstract

In the *matrix completion problem* the aim is to recover an unknown real matrix from a subset of its entries. This problem comes up in many application areas, and has received a great deal of attention in the context of the *netflix* prize.

A central approach to this problem is to output a matrix of lowest possible complexity (e.g. rank or trace norm) that agrees with the partially specified matrix. The performance of this approach under the assumption that the revealed entries are sampled randomly has received considerable attention (e.g. [1, 2, 3, 4, 5, 6, 7, 8]). In practice, often the set of revealed entries is not chosen at random and these results do not apply. We are therefore left with no guarantees on the performance of the algorithm we are using.

We present a means to obtain performance guarantees with respect to any set of initial observations. The first step remains the same: find a matrix of lowest possible complexity that agrees with the partially specified matrix. We give a new way to interpret the output of this algorithm by next finding a probability distribution over the non-revealed entries with respect to which a bound on the generalization error can be proven. The more *complex* the set of revealed entries according to a certain measure, the better the bound on the generalization error.

## 1 Introduction

In the *matrix completion problem* we observe a subset of the entries of a target matrix $Y$, and our aim is to retrieve the rest of the matrix. Obviously some restriction on the target matrix $Y$ is unavoidable as otherwise it is impossible to retrieve even one missing entry; usually, it is assumed that $Y$ is generated in a way so as to have low complexity according to a measure such as matrix rank.

A common scheme for the matrix completion problem is to select a matrix $X$ that minimizes some combination of the *complexity* of $X$ and the *distance* between $X$ and $Y$ on the observed part. In particular, one can demand that $X$ agrees with $Y$ on the observed initial sample (i.e. the *distance* between $X$ and $Y$ on the observed part is zero). This general algorithm is described in Figure 1, and we refer to it as Alg1. It outputs a matrix with minimal complexity that agrees with $Y$ on the initial sample $S$. The complexity measure can be rank, or a norm to serve as an efficiently computable proxy for the rank such as the trace norm or $\gamma_2$ norm. When we wish to mention which complexity measure is used we write it explicitly, e.g. $\mathrm{Alg1}(\gamma_2)$. Our framework is suitable using any norm satisfying few simple conditions described in the sequel.

The performance of Alg1 under the assumption that the initial subset is picked at random is well understood [1, 2, 3, 4, 5, 6, 7, 8]. This line of research can be divided into two parts. One line of research [5, 6, 4] studies conditions under which $\mathrm{Alg1}(\mathrm{Tr})$ retrieves the matrix *exactly* [1]. They

define what they call an *incoherence property* which quantifies how spread the singular vectors of $Y$ are. The exact definition of the incoherence property varies in different results. It is then proved that if there are enough samples relative to the rank of $Y$ and its incoherence property, then $\mathrm{Alg1(Tr)}$ retrieves the matrix $Y$ exactly with high probability, assuming the samples are chosen uniformly at random. Note that in this line of research the trace norm is used as the complexity measure in the algorithm. It is not clear how to prove similar results with the $\gamma_2$ norm.

Candes and Recht [5] observed that it is impossible to reconstruct a matrix that has only one entry equal to 1 and zeros everywhere else, unless most of its entries are observed. Thus, exact matrix completion must assume some special property of the target matrix $Y$. In a second line of research, general results are proved regarding the performance of Alg1. These results are weaker in that they do not prove exact recovery, but rather bounds on the distance between the output matrix $X$ and $Y$. But these results apply for every matrix $Y$, they can be generalized for non-uniform probability distributions, and also apply when the complexity measure is the $\gamma_2$ norm. These results take the following form:

**Theorem 1 ([2])** *Let $Y$ be an $n \times n$ real matrix, and $P$ a probability distribution on pairs $(i, j) \in [n]^2$. Choose a sample $S$ of $|S| > n \log n$ entries according to $P$. Then, with probability at least $1 - 2^{-n/2}$ over the sample selection, the following holds:*

$$\sum_{i,j} P_{ij} |X_{ij} - Y_{ij}| \leq c\gamma_2(X) \sqrt{\frac{n}{|S|}}.$$

*Where $X$ is the output of the algorithm with sample $S$, and $c$ is a universal constant.*

In practice, the assumption that the sample is random is not always valid. Sometimes the subset we see reflects our partial knowledge which is not random at all. What can we say about the output of the algorithm in this case? The analysis of random samples does not help us here, because these proofs do not reveal the structure that makes generalization possible. In order to answer this question we need to understand what properties of a sample enable generalization.

A first step in this direction was taken in [9] where the initial subset was chosen deterministically as the set of edges of a good expander (more generally, a good sparsifier). Deterministic guarantees were proved for the algorithm in this case, that resemble the guarantees proved for random sampling. For example:

**Theorem 2** *[9] Let $S$ be the set of edges of a $d$-regular graph with second eigenvalue [2] bound $\lambda$. For every $n \times n$ real matrix $Y$, if $X$ is the output of Alg1 with initial subset $S$, then*

$$\frac{1}{n^2} \sum_{i,j} (X_{ij} - Y_{ij})^2 \quad \leq \quad c\gamma_2(Y)^2 \frac{\lambda}{d},$$

*where $c$ is a small universal constant.*

Recall that $d$-regular graphs with $\lambda = O(\sqrt{d})$ can be constructed in linear time using e.g. the well-known LPS Ramanujan graphs [10].

This theorem was also generalized to bound the error with respect to any probability distribution. Instead of expanders, sparsifiers were used to select the entries to observe for this result.

**Theorem 3** *[9] Let $P$ be a probability distribution on pairs $(i, j) \in [n]^2$, and $d > 1$. There is an efficiently constructible set $S \subset [n]^2$ of size at most $dn$, such that for every $n \times n$ real target matrix $Y$, if $X$ is the output of our algorithm with initial subset $S$, then*

$$\sum_{i,j} P_{ij} (X_{ij} - Y_{ij})^2 \quad \leq \quad c\gamma_2(Y)^2 \frac{1}{\sqrt{d}}.$$

The results in [9] still do not answer the practical question of how to reconstruct a matrix from an arbitrary sample. In this paper we continue the work started in [9], and give a simple and general answer to this second question.

We extend the results of [9] in several ways:

1. We upper bound the generalization error of Alg1 given any set of initial observations. This bound depends on properties of the set of observed entries.

2. We show there is a probability distribution outside of the observed entries such that the generalization error under this distribution is bounded in terms of the *complexity* of the observed entries, under a certain complexity measure.

3. The results hold not only for $\gamma_2$ but also for the trace norm, and in fact any norm satisfying a few basic properties.

## 2   Preliminaries

Here we introduce some of the matrix notation and norms that we will be using. For matrices $A, B$ of the same size, let $A \circ B$ denote the Hadamard or entrywise product of $A$ and $B$. For a $m$-by-$n$ matrix $A$ with $m \geq n$ let $\sigma_1(A) \geq \cdots \geq \sigma_n(A)$ denote the singular values of $A$. The trace norm, denoted $\|A\|_{tr}$, is the $\ell_1$ norm of the vector of singular values, and the Frobenius norm, denoted $\|A\|_F$, is the $\ell_2$ norm of the vector of singular values.

As the rank of a matrix is equal to the number of non-zero singular values, it follows from the Cauchy-Schwarz inequality that

$$\frac{\|A\|_{tr}^2}{\|A\|_F^2} \leq \text{rk}(A) \ . \tag{1}$$

This inequality motivates the use of the trace norm as a proxy for rank in rank minimization problems. A problem with the bound of (1) as a complexity measure is that it is not monotone—the bound can be larger on a submatrix of $A$ than on $A$ itself. As taking the Hadamard product of a matrix with a rank one matrix does not increase its rank, a way to fix this problem is to consider instead:

$$\max_{\substack{u,v \\ \|u\|=\|v\|=1}} \frac{\|A \circ vu^T\|_{tr}^2}{\|A \circ vu^T\|_F^2} \leq \text{rk}(A) \ .$$

When $A$ is a sign matrix, this bound simplifies nicely—for then, $\|A \circ vu^T\|_F = \|u\|\|v\| = 1$, and we are left with

$$\max_{\substack{u,v \\ \|u\|=\|v\|=1}} \|A \circ vu^T\|_{tr}^2 \leq \text{rk}(A) \ .$$

This motivates the definition of the $\gamma_2$ norm.

**Definition 4** *Let $A$ be a $n$-by-$n$ matrix. Then*

$$\gamma_2(A) = \max_{\substack{u,v \\ \|u\|=\|v\|=1}} \|A \circ vu^T\|_{tr} \ .$$

We will also make use of the dual norms of the trace and $\gamma_2$ norms. Recall that in general for a norm $\Phi(A)$ the dual norm $\Phi^*$ is defined as

$$\Phi^*(A) = \max_B \frac{\langle A, B \rangle}{\Phi(B)}$$

Notice that this means that

$$\langle A, B \rangle \leq \Phi^*(A)\Phi(B) \ . \tag{2}$$

The dual of the trace norm is $\| \cdot \|$ the operator norm from $\ell_2$ to $\ell_2$, also known as the spectral norm. The dual of the $\gamma_2$ norm looks as follows.

**Definition 5**

$$\gamma_2^*(A) = \min_{\substack{X,Y \\ X^TY=A}} \frac{1}{2} \left( \|X\|_F^2 + \|Y\|_F^2 \right)$$

$$= \min_{\substack{X,Y \\ X^TY=A}} \|X\|_F \|Y\|_F \ ,$$

*where the min is taken over $X, Y$ with orthogonal columns.*

Finally, we will make use of the approximate $\gamma_2$ norm. This is the minimum of the $\gamma_2$ norm over all matrices which approximate the target matrix in some sense. The particular version we will need is denoted $\gamma_2^{0,\infty}$ and is defined as follows.

**Definition 6** *Let $S \in \{0,1\}^{m \times n}$ be a boolean matrix. Let $\bar{S}$ denote the complement of $S$, that is $\bar{S} = J - S$ where $J$ is the all ones matrix. Then*

$$\gamma_2^{0,\infty}(S) = \min_T \{\gamma_2(T) : T \circ S \geq S, T \circ \bar{S} = 0\}$$

In words, $\gamma_2^{0,\infty}(S)$ is the minimum $\gamma_2$ norm of a matrix $T$ which is 0 whenever $S$ is zero, and at least 1 whenever $S$ is 1. This can be thought of as a "one-sided error" version of the more familiar $\gamma_2^{\infty}$ norm of a sign matrix, which is the minimum $\gamma_2$ norm of a matrix which agrees in sign with the target matrix and has all entries of magnitude at least 1. The $\gamma_2^{\infty}$ bound is also known to be equal to the margin complexity [11].

## 3   The algorithm

Let $S \subset [m] \times [n]$ be a subset of entries, representing our partial knowledge. We can always run Alg1 and get an output matrix $X$. What we need in order to make intelligent use of $X$ is a way to measure the distance between $X$ and $Y$. Our first observation is that although $Y$ is not known, it is possible to bound the distance between $X$ and $Y$. This result is stated in the following theorem which generalizes Theorems (2) and (4) of [9] [3]:

**Theorem 7** *Fix a set of entries $S \subset [m] \times [n]$. Let $P$ be a probability distribution on pairs $(i, j) \in [m] \times [n]$, such that there exists a real matrix $Q$ satisfying*

1. *$Q_{ij} = 0$ when $(i, j) \notin S$.*

2. *$\gamma_2^*(P - Q) \leq \Lambda$*

*Then for every $m \times n$ real target matrix $Y$, if $X$ is the output of our algorithm with initial subset $S$, it holds that*

$$\sum_{i,j} P_{ij}(X_{ij} - Y_{ij})^2 \quad \leq \quad 4\Lambda\gamma_2(Y)^2 \ .$$

Theorem 7 says that $\gamma_2^*(P - Q)$ determines, at least to some extent, the expected distance between $X$ and $Y$ with respect to $P$.

This gives us a way to measure the quality of the output of Alg1 for any set $S$ of initial observations. Namely, we can do the following:

1. Choose a probability distribution $P$ on the entries of the matrix.

2. Find a real matrix $Q$ such that $Q_{ij} = 0$ when $(i, j) \notin S$, and $\gamma_2^*(P - Q)$ is minimal.

3. Output the minimal value $\Lambda$.

We then know, using Theorem 7, that the expected square distance between $X$ and $Y$ can be bounded in terms of $\Lambda$ and the complexity of $Y$.

Obviously, the choice of $P$ makes a big difference. For example if the set of initial observations is contained in a submatrix we cannot expect $X$ to be close to $Y$ outside this submatrix. In such cases it makes sense to restrict $P$ to the submatrix containing $S$.

One approach to find a distribution for which we can expect to be close on the unseen entries is to optimize over probability distributions $P$ such that Theorem 7 gives the best bound. Since $\gamma_2^*$ can be expressed as the optimum of semidefinite program, we can find in polynomial time a probability distribution $P$ and a weight function $Q$ on $S$ such that $\gamma_2^*(P - Q)$ is minimizd. Thus, instead of trying different parameters, we can find a probability distribution for which we can prove optimal

1. Input: a subset $S \subset [n]^2$ and the value of Y on $S$.
2. Output: a matrix $X$ of smallest possible CC(X) under the condition that $X_{ij} = Y_{ij}$ for all $(i, j) \in S$.
---

Figure 1: Algorithm $\mathrm{Alg1}(CC)$

guarantees using Theorem 7. The second algorithm we suggest does exactly that. We refer to this algorithm as Alg2, or $\mathrm{Alg2}(CC)$ if we wish to state the complexity measure that is used.

For $\mathrm{Alg2}(\gamma_2)$, we do the following: Minimize $\gamma_2^*(P - Q)$ over all $m \times n$ matrices $Q$ and $P$ such that:

1. $Q_{ij} = 0$ for $(i, j) \notin S$.

2. $P_{ij} = 0$ for $(i, j) \in S$.

3. $\sum_{i,j} P_{ij} = 1$.

Globally, our algorithm for matrix completion therefore works in two phases. We first use Alg1 to get an output matrix $X$, and then use Alg2 in order to find optimal guarantees regarding the distance between $X$ and $Y$. The generalization error bounds for this algorithm are proved in Section 4.

### 3.1 Using a general norm

In our description of Alg2 above we have used the norm $\gamma_2$. The same idea works for any norm $\Phi$ satisfying the property $\Phi(A \circ A) \leq \Phi(A)^2$. Moreover, if the dual norm can be computed efficiently via a linear or semidefinite program, then the optimal distribution $P$ for the bound can be found efficiently as well.

For example for the trace norm the algorithm becomes: Given the sample $S$ run $\mathrm{Alg1}(\| \cdot \|_{tr})$ and get an output matrix $X$. The second part of the algorithm is: Minimize $\|P - Q\|$ over all $m \times n$ matrices $Q$ and $P$ such that:

1. $Q_{ij} = 0$ for $(i, j) \notin S$.

2. $P_{ij} = 0$ for $(i, j) \in S$.

3. $\sum_{i,j} P_{ij} = 1$.

Denote by $\Lambda$ the optimal value of the above program, and by $P$ the optimal probability distribution. Then analogously to Theorem 7, we have

$$\sum_{i,j} P_{ij}(X_{ij} - Y_{ij})^2 \quad \leq \quad 4\Lambda \|Y\|_{tr}^2 \ .$$

Both of these results will follow from a more general theorem which we show in the next section.

## 4 Generalization bounds

Here we show a more general theorem which will imply Theorem 7.

**Theorem 8** *Let $\Phi$ be a norm and $\Phi^*$ its dual norm. Suppose that $\Phi(A \circ A) \leq \Phi(A)^2$ for any matrix A.*

*Fix a set of indices $S \subset [m] \times [n]$. Let $P$ be a probability distribution on pairs $(i, j) \in [m] \times [n]$, such that there exists a real matrix $Q$ satisfying*

1. *$Q_{ij} = 0$ when $(i, j) \notin S$.*

2. *$\Phi^*(P - Q) \leq \Lambda$*

*Then for every $m \times n$ real target matrix $Y$, if $X$ is the output of algorithm $\mathrm{Alg1}(\Phi)$ with initial subset $S$, it holds that*

$$\sum_{i,j} P_{ij}(X_{ij} - Y_{ij})^2 \quad \leq \quad 4\Phi(Y)^2\Lambda.$$

**Proof** Let $R$ be the matrix where $R_{ij} = (X_{ij} - Y_{ij})^2$. By assumption $\Phi^*(P - Q) \leq \Lambda$ thus by (2)

$$\langle P - Q, R\rangle \leq \Lambda\Phi(R) \ .$$

Now let us focus on $\Phi(R)$. As $R = (X - Y) \circ (X - Y)$ by the assumption on $\Phi$ we have

$$\Phi(R) \leq \Phi(X - Y)^2 \leq (\Phi(X) + \Phi(Y))^2 \ .$$

Now by definition of $\mathrm{Alg1}(\Phi)$ we have $\Phi(X) \leq \Phi(Y)$, thus $\Phi(R) \leq 4\Phi(Y)^2$. Also, by definition of the algorithm $R_{ij} = 0$ for $(i, j) \in S$, and $Q_{ij}$ equals zero outside of $S$, which implies that $\sum_{i,j} Q_{ij}R_{ij} = 0$. We conclude that

$$\sum_{i,j} P_{ij}(X_{ij} - Y_{ij})^2 \leq 4\Lambda\Phi(Y)^2.$$

■

Both the trace norm and $\gamma_2$ norm satisfy the condition of the theorem as they are multiplicative under tensor product.

## 5    Analyzing the error bound

We now look more closely at the minimal value of the parameter $\Lambda$ from Theorem 7. The optimal value of $\Lambda$ depends only on the set of observed indices $S$. For a set of indices $S \subset [m] \times [n]$ let $\bar{S}$ be its complement.

Given samples $S$ we want to find $P, Q$ so as to minimize $\gamma_2^*(P - Q)$ such that $P$ is a probability distribution over $\bar{S}$ and $Q$ has support in $S$. We can express this as a semidefinite program

$$\begin{aligned}
\Lambda = \underset{\alpha, P, Q}{\text{minimize}} \quad & \frac{1}{2}\mathrm{Tr}(\alpha) \\
\text{subject to} \quad & \alpha - (\hat{P} - \hat{Q}) \succeq 0 \\
& P \geq 0 \\
& \langle P, \bar{S}\rangle = 1 \\
& \langle Q, S\rangle = Q.
\end{aligned}$$

Here

$$\hat{P} = \begin{bmatrix} 0 & P \\ P^T & 0 \end{bmatrix}$$

is the "bipartite" version of $P$, and similarly for $\hat{Q}$.

Taking the dual of this program we find

$$\begin{aligned}
1/\Lambda = \underset{A}{\text{minimize}} \quad & \gamma_2(A) \\
\text{subject to} \quad & A \geq \bar{S} \\
& A \circ \bar{S} = A
\end{aligned}$$

In words, this says that that $\frac{1}{\Lambda}$ is equal to the minimum $\gamma_2$ norm of a matrix that is zero on all entries in $S$ and at least 1 on all entries in $\bar{S}$. Thus $\Lambda = 1/\gamma_2^{0,\infty}(\bar{S})$ (recall Definition 6). This says that the more complex the set of unobserved entries $\bar{S}$ according to the measure $\gamma_2^{0,\infty}$, the smaller the value of $\Lambda$. Note that in particular, if we consider the sign matrix $\bar{S} - S$ then $\gamma_2^{0,\infty}(\bar{S}) \geq (\gamma_2^\infty(\bar{S} - S) - 1)/2$ is lower bounded by the margin complexity of $S - \bar{S}$.

## Footnotes

[1]There are other papers studying exact matrix completion, e.g. [7].

[2] The eigenvalues are eigenvalues of the adjacency matrix of the graph.

[3]Here we state the result for $\gamma_2$. See Section 4 for the corresponding result for the trace norm as well.

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
