[Reviews · NeurIPS 2013]

Submitted by Assigned_Reviewer_4

Existing bounds for matrix completion from the trace and \gamma_2 norms require that the observed sample positions be chosen either randomly or from a specific deterministic strategy. This paper, in contrast, considers the matrix completion problem when the set of observed samples is arbitrary. A bound is derived on the error of the reconstructed unobserved entries in terms of a probability distribution over those entries which must satisfy certain conditions, and the error bound improves with the complexity of the observed sample set. The probability distribution involved in the bound can be optimized in order to provide the tightest reconstruction guarantee.

Quality: This paper considers an important problem and derives a novel bound that applies to matrix reconstruction using any norm that satisfies a certain property (including the trace and \ell_2 norms). The bounds in this paper can be somewhat difficult to interpret at first glance, but the author(s) do a satisfactory job in helping the reader interpret the results. The fact that the probability distribution involved in the bound can be optimized is a plus.

Clarity: This paper is clearly written, and the proof of the main result is surprisingly concise.

Originality: Existing bounds for matrix completion from the trace and \gamma_2 norms require that the observed sample positions be chosen either randomly or from a specific deterministic strategy. This paper is original in allowing the sample positions to be chosen arbitrarily.

Significance: Extending the literature on matrix completion guarantees to allow for arbitrary sample positions is a significant contribution. The bounds themselves in this paper can be somewhat difficult to interpret, and it is conceivable that other researchers may derive bounds of different flavors in the future. However, this paper seems like a strong first step in a new direction.

Some minor comments: Line 045: Figure 3 should be Figure 1? Line 070-071: missing a minus sign in exponent? Line 100: do the author(s) really do anything to "enhance" Alg1 in this paper? Line 185: the result depends on \gamma_2(Y). Could the author(s) mention whether it is possible to relate this to \gamma_2(X)? Line 195: should \lambda be \Lambda? Lines 211-213 and 235-238: sometimes symbols are lowercase, sometimes uppercase. Is this intentional?

UPDATE: Thanks to the authors for their notes in the rebuttal period.
Summary: This paper provides bounds on matrix completion when the set of observed samples is arbitrary. The bounds themselves in this paper can be somewhat difficult to interpret, but this paper seems like a strong first step in a new direction.

Submitted by Assigned_Reviewer_5

The authors provide a novel norm minimization algorithm for low-rank matrix completion for a fixed, deterministic set of observed entries, Alg2, which can be seen as a generalization of trace norm minimization. They prove reconstruction guarantees in the noise-free case for arbitrary norms, including the popular trace norm, which describe the reconstruction error in terms of the complexity of the set of observed entries.

The paper is well-written with minor typos (mostly, formatting such as the single section 3.1, or line breaks in weird places), and the results appear to be correct.

The results are novel and original; they seem to provide a break-through in the understanding of matrix completion. Since any assumption that the positions of the observations are random of special kind (e.g., Bernoulli, uniform) are a-posteriori uncheckable, algorithms and theoretical guarantees for fixed sets of observations are the only way of taking the structure of the given observations into account.

While the theoretical results are in my opinion very nice and strong enough to stand for themselves, the paper does not validate Alg2 on synthetic or real data, nor does it relate the bounds to known sampling scenarios (e.g. the Candès/Tao results). Therefore it is not easy to judge the impact and the effectivity of the results.
Summary: A paper providing a novel and original algorithm for low-rank matrix completion with exact reconstruction guarantees for any fixed set of observations, likely to have seminal impact.

Submitted by Assigned_Reviewer_6

Summary:
The paper discusses bounds on matrix completion problem under the assumption that the entries of the matrix are no necessarily sampled at random. This is done by optimizing the probability distributions on pairs of entries and using that to compute the error bounds according to complexity of the observed entries. The authors show that these results can be extended to other standard matrix norms.

Comments:
1) While the authors mentions that Theorem 6 is a generalization of theorems (2) and (4) in [9], theorem (4) is not explicitly stated. It might be a good idea to explicitly state it for the sake of completion.

2) While the bounds in the paper seem valid and very useful, it is hard to interpret the effectiveness of these bounds. It would be nice to have empirical studies comparing the bounds in this paper to previous works.
Summary: The bound in the paper is novel. However, empirical/simulation studies on the effectiveness of these bounds would be nice. It is also a good idea to explicitly state both theorems in [9] so readers will have a better understanding on how much was extended from previous work.
Author Feedback

Author rebuttal: Thank you for the reviews. The reviews show an accurate understanding of the paper and give a fair judgement of its merits. We will implement the suggested changes of Assigned_Reviewer_4 in the final version, and also add a statement of Theorem 4 from [9] as suggested by Assigned_Reviewer_6. Finally, we agree that experimental data is very interesting. We are working on implementations of the algorithm and benchmarking its performance, and plan to present this in future work.